



# 1 Biogenic halocarbons from the Peruvian upwelling region
# 2 as tropospheric halogen source

**Helmke Hepach[1], Birgit Quack[1], Susann Tegtmeier[1], Anja Engel[1], Astrid**
**Bracher[2], Steffen Fuhlbrügge[1], Luisa Galgani[1], Elliot Atlas[3], Johannes**
**Lampel[4,5], Udo Frieß[4], and Kirstin Krüger[6]**
Correspondence to: H. Hepach (hhepach@geomar.de)
[1] GEOMAR Helmholtz Centre for Ocean Research Kiel, Germany
[2] Alfred Wegener Institute (AWI), Helmholtz Centre for Polar and Marine Research
Bremerhaven, Germany and Institute of Environmental Physics, University of Bremen,
Germany
[3] Rosenstiel School of Marine and Atmospheric Science (RSMAS), University of Miami,
USA
[4] Institute of Environmental Physics, University of Heidelberg, Germany
[5] now at Max Planck Institute for Chemistry, Mainz, Germany
[6] Department of Geosciences, University of Oslo, Oslo, Norway





**Abstract**
Halocarbons, halogenated short-chained hydrocarbons, are produced naturally in the oceans
by biological and chemical processes. They are emitted from surface seawater into the
atmosphere, where they take part in numerous chemical processes such as ozone destruction
and the oxidation of mercury and dimethyl sulfide. Here we present oceanic and atmospheric
halocarbon data for the Peruvian upwelling obtained during the M91 cruise onboard the
research vessel *Meteor* in December 2012. Surface waters during the cruise were
characterized by moderate concentrations of bromoform ($CHBr_3$) and dibromomethane
($CH_2Br_2$) correlating with diatom biomass derived from marker pigment concentrations,
which suggests this phytoplankton group as likely source. Concentrations measured for the
iodinated compounds methyl iodide ($CH_3I$) of up to 35.4 pmol $L^{-1}$, chloroiodomethane
($CH_2ClI$) of up to 58.1 pmol $L^{-1}$ and diiodomethane ($CH_2I_2$) of up to 32.4 pmol $L^{-1}$ in water
samples were much higher than previously reported for the tropical Atlantic upwelling
systems. Iodocarbons also correlated with the diatom biomass and even more significantly
with dissolved organic matter (DOM) components measured in the surface water. Our results
suggest a biological source of these compounds as significant driving factor for the observed
large iodocarbon concentrations. Elevated atmospheric mixing ratios of $CH_3I$ (up to 3.2 ppt),
$CH_2ClI$ (up to 2.5 ppt) and $CH_2I_2$ (3.3 ppt) above the upwelling were correlated with seawater
concentrations and high sea-to-air fluxes. The enhanced iodocarbon production in the
Peruvian upwelling contributed significantly to tropospheric iodine levels.
**1    Introduction**
Brominated and iodinated short-lived organic compounds (halocarbons) from the oceans
contribute to tropospheric and stratospheric chemistry (von Glasow et al., 2004; Saiz-Lopez et
al., 2012b; Carpenter and Reimann, 2014). They are significant carriers of iodine and bromine
into the marine atmospheric boundary layer (Salawitch, 2006; Jones et al., 2010; Yokouchi et
al., 2011; Saiz-Lopez et al., 2012b), where they and their degradation products may also be
involved in aerosol and ultra-fine particle formation (O'Dowd et al., 2002; Burkholder et al.,
2004). Furthermore, the short-lived organic compounds are a source for bromine and iodine in
the free troposphere, and thus play an important role for ozone chemistry and other processes
such as the oxidation of several atmospheric constituents (Saiz-Lopez et al., 2012a).
Numerous modelling studies over the last years have shown that brominated short-lived
compounds and their degradation products can be entrained into the stratosphere and enhance





the halogen-driven ozone destruction (Carpenter and Reimann, 2014; Hossaini et al., 2015).
Recently, it was suggested that also oceanic iodine in organic or inorganic form can contribute
to the stratospheric halogen loading, however only in small amounts due to its strong
degradation (Tegtmeier et al., 2013; Saiz-Lopez et al., 2015).
While different source and sink processes determine the distribution of halocarbons in the
oceanic surface water, the underlying mechanisms are largely unresolved. Biological activity
plays a role for the production of bromoform ($CH Br_3$), dibromomethane ($CH_2Br_2$), methyl
iodide ($CH_3I$), chloroiodomethane ($CH_2ClI$) and diiodomethane ($CH_2I_2$) (Gschwend et al.,
1985; Tokarczyk and Moore, 1994; Moore et al., 1996), while $CH_3I$ also originates from
photochemical reactions with dissolved organic matter (DOM) (Moore and Zafiriou, 1994;
Bell et al., 2002; Shi et al., 2014).
Biologically mediated halogenation of DOM (Lin and Manley, 2012; Liu et al., 2015) and
bromination of compounds such as β-diketones via enzymes such as bromoperoxidase (BPO)
within or outside the algal cells (Theiler et al., 1978) are among the potentially important
production processes of $CH Br_3$ and $CH_2Br_2$. Air-sea gas exchange into the atmosphere is the
most important sink for both compounds (Quack and Wallace, 2003; Hepach et al., 2015).
The biological formation of $CH_3I$ has been investigated during laboratory and field studies
(Scarratt and Moore, 1998; Amachi et al., 2001; Fuse et al., 2003; Smythe-Wright et al., 2006;
Brownell et al., 2010; Hughes et al., 2011) identifying phyto- and bacterioplankton as
producers, revealing large variability in biological production rates. However, biogeochemical
modelling studies suggest that photochemistry may be more important for global $CH_3I$
production (Stemmler et al., 2014). Due to their much shorter lifetime in surface water and the
atmosphere, fewer studies investigated production processes of $CH_2I_2$ and $CH_2ClI$. $CH_2I_2$ has
been suggested to be produced both by phytoplankton (Moore et al., 1996) and bacteria (Fuse
et al., 2003; Amachi, 2008). The main source for $CH_2ClI$ is likely its production during the
photolysis of $CH_2I_2$ with a yield of 35 % based on a laboratory study (Jones and Carpenter,
2005). $CH_2ClI$ has also been detected in phytoplankton cultures (Tokarczyk and Moore, 1994)
where it may originate from direct production or also from $CH_2I_2$ conversion. The main sink
for both $CH_2I_2$ and $CH_2ClI$ is their photolytic destruction in the surface ocean resulting in
lifetimes of less than 10 min ($CH_2I_2$) and 9 h ($CH_2ClI$), respectively, in the tropical ocean
(Jones and Carpenter, 2005; Martino et al., 2006). Other sinks for these three iodocarbons are
air-sea gas exchange and chloride substitution. The latter may play an important role for $CH_3I$
in low latitudes at low wind speeds (Zafiriou, 1975; Jones and Carpenter, 2007).



Oceanic measurements of natural halocarbons are sparse (Ziska et al., 2013), but reveal that
especially tropical and subtropical upwelling systems are potentially important source regions
(Quack et al., 2007a; Raimund et al., 2011). Previously observed high tropospheric iodine
monoxide (IO) levels in the tropical East Pacific have been related to short-lived iodinated
compounds in surface waters (Schönhardt et al., 2008; Dix et al., 2013). However, iodocarbon
fluxes have not been considered high enough to explain observed IO concentrations (Jones et
al., 2010; Mahajan et al., 2010; Grossmann et al., 2013; Lawler et al., 2014), and recent global
modelling studies suggested abiotic sources contributing on average about 75 % to the IO
budget (Prados-Roman et al., 2015). Such abiotic sources could be emissions of hypoiodous
acid (HOI) and molecular iodine ($I_2$) as recently confirmed by a laboratory study (Carpenter et
al., 2013).
This paper characterizes the Peruvian upwelling region between 5.0° S, 82.0° W and 16.2° S,
76.8° W with regard to the two brominated compounds $CHBr_3$ and $CH_2Br_2$ and the iodinated
compounds $CH_3I$, $CH_2ClI$ and $CH_2I_2$ in water and atmosphere. The latter two compounds
were measured for the first time in this region. Possible oceanic sources based on the analysis
of phytoplankton species composition and different DOM components were evaluated and
identified. Sea-to-air fluxes of these halogenated compounds were derived and their
contribution to the tropospheric iodine loading above the tropical East Pacific by combining
halocarbon and IO measurements and model calculations were estimated.

## 21   **2   Methods**

The M91 cruise of the RV *Meteor* from December 1 to 26, 2012 investigated the surface
ocean and atmosphere of the Peruvian upwelling region (Bange, 2013). From the
northernmost location of the cruise at 5.0° S and 82.0° W, the ship moved to the southernmost
position at 16.2° S and 76.8° W with several transects perpendicular to the coast, alternating
between open ocean and coastal upwelling (Fig. 1). All underway measurements were taken
from a continuously operating pump in the ship's hydrographic shaft from a depth of 6.8 m.
Sea surface temperature (SST) and sea surface salinity (SSS) were measured continuously
with a SeaCAT thermosalinograph from Seabird Electronics (SBE).
Deep samples were taken from 4 to 10 depths between 1 and 2000 m from 12 L Niskin bottles
attached to a 24-bottle-rosette sampler equipped with a CTD and an oxygen sensor from SBE.
Halocarbon samples were collected at 24 of the total 98 casts. The uppermost sample from the
depth profiles (between 1 and 10 m) was included in the surface water measurements.



## 2.1  Analysis of halocarbon samples
Halocarbon samples were taken three hourly from sea surface water and air. Surface water
samples were analyzed on board with a purge and trap system attached to a GC-MS
(combined gas chromatography and mass spectrometry) described in more detail in Hepach et
al. (2014). The depth profile samples were analyzed with a similar setup: a purge and trap
system was attached to a GC equipped with an ECD (electron capture detector). The precision
of the measurements lay within 10 % for all five halocarbons determined from duplicates and
both systems were calibrated using the same liquid standards in methanol. Halocarbon
measurements in seawater started only on December 9 due to set up problems. Atmospheric
halocarbon samples were taken on the monkey deck at a height of 20 m using a metal bellows
pump from December 1, and were analyzed at the Rosenstiel School of Marine and
Atmospheric Science (RSMAS) as described in Schauffler et al. (1998). For further details of
atmospheric measurements see Fuhlbrügge et al. (2015a). Quantification was achieved using
the NOAA standard SX3573 from GEOMAR.
## 2.2  Biological parameters
Phytoplankton composition was derived from pigment concentrations.  Samples were taken in
parallel with the halocarbon samples in the sea surface and up to six samples in depths
between 3 and 200 m. Water was filtered with GF/F filters, which were stored at -80 °C until
analysis after shock-freezing in liquid nitrogen. Pigments as described in Taylor et al. (2011)
were analyzed using a HPLC technique according to Barlow et al. (1997). We used the
diagnostic pigment analysis by Vidussi et al. (2001), subsequently refined by Uitz et al.
(2006) by introducing pigment specific weight coefficients, to determine the chlorophyll *a*
(Chl *a*) concentration of seven groups of phytoplankton which are assumed to build up the
entire phytoplankton community in ocean waters. Identified phytoplankton groups include
diatoms, chlorophytes, dinoflagellates, haptophytes, cyanobacteria, cryptophytes and
chrysophytes. Total chlorophyll a (TChl *a*) concentrations were calculated from the sum of
the pigment concentrations of monovinyl Chl *a*, divinyl Chl *a* and chlorophyllide *a*.
Samples for the identification of DOM components were taken at 37 stations from a rubber
boat from subsurface water at approximately 20 cm (further called "subsurface"). All samples
were processed onboard and analyzed back in the home laboratory. Samples were analyzed
for dissolved and total organic carbon (DOC and TOC), total dissolved nitrogen (TDN), total
nitrogen (TN), total, dissolved and particulate high molecular weight (HMW, >1kDa)



combined carbohydrates (TCCHO, DCCHO and PCCHO) applying a high-temperature
catalytic oxidation method using a TOC analyzer (TOC-$V_{CSH}$) from Shimadzu, as well as
total, dissolved and particulate combined HMW uronic acids (TURA, DURA and PURA), i.e.
galacturonic acid and glucuronic acid. These were analyzed by High Performance Anion
Exchange Chromatography coupled with Pulsed Amperometric Detection (HPAEC-PAD)
after Engel and Händel (2011). For a more detailed description of both the sampling method
and analysis see Engel and Galgani (2015).
**2.3   Correlation analysis**
Correlation analyses between all halocarbons, biological proxies and ambient parameters were
carried out using Matlab® for all collocated surface and depth samples. All datasets were
tested for normal distribution using the Lilliefors-test. Since most of the data were not
distributed normally, Spearman's rank correlation (hereinafter called $r_s$) was used. All
correlations with a significance level of smaller than 5 % ($p < 0.05$) were regarded as
significant.
**2.4   Calculation of sea-to-air fluxes**
Sea-to-air fluxes $F$ of halocarbons were calculated according to equation 1 with $k_w$ as the gas
exchange coefficient parameterized according to Nightingale et al. (2000), $c_w$ the water
concentrations from the halocarbon underway measurements, $c_{atm}$ from the simultaneous
atmospheric measurements and $H$ as the Henry's law constant to derive the equilibrium
concentration.
$$F = k_w \cdot (c_w - \frac{c_{atm}}{H})$$  (1)
The gas exchange coefficient usually applied to derive carbon dioxide fluxes was adjusted for
halocarbons using Schmidt number corrections as calculated in Quack and Wallace (2003),
and Henry's law coefficients as reported for each of the compounds by Moore et al. (1995)
were applied. Wind speed and air pressure were averaged to 10 min intervals for the
calculation of the instantaneous fluxes.
**2.5   FLEXPART simulations of tropospheric iodine**
The atmospheric transport of the iodocarbons from the oceanic surface into the Marine
Atmospheric Boundary Layer (MABL) was simulated with the Lagrangian particle dispersion





model FLEXPART (Stohl et al., 2005) which has been used extensively in studies of long-
range and mesoscale transport (Stohl and Trickl, 1999). FLEXPART is an off-line model
driven by external meteorological fields. It includes parameterizations for moist convection,
turbulence in the boundary layer, dry deposition, scavenging, and the simulation of chemical
decay. We simulate trajectories of a multitude of air parcels describing transport and chemical
decay of the emitted oceanic iodocarbons. For each data point of the observed sea-to-air flux,
100000 air parcels were released over the duration of the M91 cruise from a 0.1° x 0.1° grid
box at the ocean surface centered at the measurement location. We used FLEXPART version
9.2 and the runs are driven by the ECMWF reanalysis product ERA-Interim (Dee et al., 2011)
given at a horizontal resolution of 1° x 1° on 60 model levels. Transport, dispersion and
convection of the air parcels are calculated from the 6-hourly fields of horizontal and vertical
wind, temperature, specific humidity, convective and large scale precipitation and others. The
chemical decay of the iodocarbons was prescribed by their atmospheric lifetime which was set
to 4 days, 9 hours and 10 min for $CH_3I$, $CH_2ClI$, and $CH_2I_2$, respectively, according to current
estimates (Jones and Carpenter, 2005; Martino et al., 2006; Carpenter and Reimann, 2014).
After degradation of the iodocarbons, the released iodine was simulated as inorganic iodine
($I_y$) tracer with a prescribed lifetime in the marine boundary layer of two days (personal
communication R. von Glasow). Thus we did not include detailed tropospheric iodine
chemistry, explicit removal of HOI, HI, $IONO_2$, and $I_xO_y$ through scavenging or
heterogeneous recycling of HOI, $IONO_2$, and $INO_2$ on aerosols (Saiz-Lopez et al., 2014). In
order to estimate the uncertainties arising from this simplification, we conducted two
additional simulations, one with a very short lifetime of one day and one with a longer
lifetime of three days. Following model simulations of halogen chemistry for air masses from
different oceanic regions in Sommariva and von Glasow (2012), IO corresponds to 20 % of
the $I_y$ budget in the marine boundary layer on a daytime average. The IO to $I_y$ ratio shows
moderate changes with daytime resulting in highest IO proportion at sunrise (~ 30 %) and
lowest IO proportion around noon (~ 15 %) (see Fig. S6 in Sommariva and von Glasow
(2012)). The ratio shows only very small variations for different air mass origins and thus the
chemical conditions such as ozone and nitrogen species concentrations. Additionally, the ratio
does not change much with altitude within the marine boundary layer. Based on the above
estimates from Sommariva and von Glasow (2012), we used the IO to $I_y$ ratio as a function of
daytime to estimate IO from $I_y$ every 3 hours. Daily averages of the IO abundance were
compared to the MAX-DOAS IO measurements on board described in the next section.





**2.6  MAX-DOAS Measurements of IO**
Multi-AXis Diffential Optical Absorption Spectroscopy (MAX-DOAS) (Hönninger, 2002;
Platt and Stutz, 2008) observations were conducted continuously at daytime from November
30 to December 25 2012 in order to quantify tropospheric abundances of IO, BrO, HCHO,
Glyoxal, $NO_2$ and HONO along the cruise track, and for aerosol profiles also of $O_4$. The
MAX-DOAS instrument and the measurement procedure are described in Grossmann et al.
(2013) and Lampel et al. (2015).
The primary quantity derived from MAX-DOAS measurements is the differential slant
column density (dSCD), which represents the difference in path-integrated concentrations
between two measurements in off-axis and zenith direction. From the MAX-DOAS
observations of $O_4$ dSCD aerosol extinction profiles to estimate the quality of visibility were
inferred using an optimal estimation approach described in Frieß et al. (2006) and Yilmaz
(2012) after applying a correction factor of 1.25 to the $O_4$ dSCDs (Clémer et al., 2010). IO
was analyzed in the spectral range from 418 – 438 nm following the settings in Lampel et al.
(2015). IO was found up to 6 times above the detection limit (twice the measurement error).
**3   The tropical East Pacific – general description and state during M91**
The tropical East Pacific is characterized by one of the strongest and most productive all-year-
prevailing eastern boundary upwelling systems of the world (Bakun and Weeks, 2008).
Temperatures drop to less than 16 °C when cold water from the Humboldt current is
transported to the surface due to Ekman transport caused by strong equatorward winds
(Tomczak and Godfrey, 2005), which is also connected with an upward transport of nutrients
(Chavez et al., 2008). As a consequence of the enhanced nutrient supply and the high solar
insolation, phytoplankton blooms, indicated by high Chl $a$ values, can be observed at the
surface especially in the boreal winter months (Echevin et al., 2008). A strong oxygen
minimum zone (OMZ) is formed due to enhanced primary production, sinking particles and
weak circulation (Karstensen et al., 2008).
Low SSTs of mean (min – max) 19.4 (15.0 – 22.4) °C and high TChl $a$ values of on average
1.80 (0.06 – 12.65) µg $L^{-1}$ (Table 1, Fig. 1) were measured during our cruise. Diatoms were
dominating the TChl $a$ concentration in the surface water with a mean of 1.66 (0.00 – 10.47)
µg Chl $a$ $L^{-1}$, followed by haptophytes (mean: 0.25 µg Chl $a$ $L^{-1}$), chlorophytes (mean: 0.19
µg Chl $a$ $L^{-1}$), cyanobacteria (mean: 0.09 µg Chl $a$ $L^{-1}$), dinoflagellates (mean: 0.08 µg Chl $a$
$L^{-1}$), cryptophytes (mean: 0.03 µg Chl $a$ $L^{-1}$), and finally chrysophytes (mean: 0.03 µg Chl $a$





L$^{-1}$). Diatoms were observed at all stations with concentrations above 0.5 µg Chl $a$ L$^{-1}$ to
contribute more than 50 % of the algal biomass. They correlated very well with TChl $a$ (Table
2) and with cryptophytes, which were elevated in very similar regions. Abundance of these
phytoplankton groups was strongly anticorrelated with SST and SSS, indicating a close
conjunction with the colder and more saline upwelling waters. Nutrients (nitrate, nitrite,
ammonium and phosphate) were also measured during the cruise (see Czeschel et al. (2015)
for further information). A weak anticorrelation of phytoplankton group with the ratio of
dissolved inorganic nitrogen and phosphate (sum of nitrate, nitrite and ammonium divided by
phosphate, DIN:DIP) (Table 2) indicated that diatoms and cryptophytes were more abundant
in aged upwelling, where nutrients were already slightly depleted or used up. The TChl $a$
maximum was generally found in the surface ocean except for four stations with overall low
TChl $a$ (< 0.5 µg L$^{-1}$) where a subsurface maximum around 30 and 50 m was identified.
All regions with SSTs below the mean of 19.4 °C are considered as upwelling in the
following sections for identifying different significant regions for halocarbon production.
Based on this criterion, four upwelling regions (I – IV) close to the coast were classified (Fig.
1). The most intense upwelling (lowest SSTs, high nutrient concentrations) appeared in the
northernmost region of the cruise track, region I, while higher TChl $a$ and lower nutrients
indicate a fully developed bloom in the southern part of the cruise (upwelling regions III and
IV). Upwelling region II was characterized by a lower DIN:DIP ratio in contrast to region I.
SSS with a mean of 34.95 (34.10 and 35.50) is lowest in upwelling region IV, which is likely
influenced by local river input such as the rivers Pisco, Cañete and Matagente, and may
explain the observed low salinities due to enhanced fresh water input in boreal winter
(Bruland et al., 2005).

## 4  Halocarbons in the surface water and depth profiles during M91

### 4.1  Halocarbon distribution in surface water

Measurements of halocarbons in the tropical East Pacific are very sparse and no data were
available for the Peruvian upwelling system before our campaign. Sea surface concentrations
of CHBr$_3$ with a mean of 6.6 (0.2 – 21.5) and of CH$_2$Br$_2$ of 4.3 (0.2 – 12.7) pmol L$^{-1}$ were
measured during M91 (Table 1, Fig. 2). These values are low in comparison to 44.7 pmol L$^{-1}$
CHBr$_3$ in tropical upwelling systems in the Atlantic, while our measurements of CH$_2$Br$_2$
compare better to these upwelling systems, from which maximum concentrations of 9.4 pmol





$L^{-1}$ were reported (Quack et al., 2007a; Carpenter et al., 2009; Hepach et al., 2014, 2015).
$CHBr_3$ (0.2– 20.7 pmol $L^{-1}$) and $CH_2Br_2$ (0.7 – 6.5 pmol $L^{-1}$) concentrations in the tropical
East Pacific open ocean and Chilean coastal waters during a cruise from Punta Arenas, Chile
to Seattle, USA in April 2010 (Liu et al., 2013) compare well to our data. Some
measurements also exist for the tropical West Pacific with on average 0.5 to 3 times the
$CHBr_3$ and 0.2 to 1 times the $CH_2Br_2$ during our cruise with the high average originating from
a campaign close to the coast with macroalgal and anthropogenic sources (Krüger and Quack,
2013; Fuhlbrügge et al., 2015b). $CHBr_3$ and $CH_2Br_2$ have been proposed to have similar
sources (Moore et al., 1996; Quack et al., 2007b). However, during our cruise, the correlation
between the two compounds was comparatively weak ($r_s$ = 0.56), consistent with the findings
of Liu et al. (2013), who ascribed the weaker correlation of these two compounds to formation
in a common ecosystem rather than to the exact same biological sources. Maxima of $CH_2Br_2$
were observed in both upwelling regions III and IV, while $CHBr_3$ was highest in the most
southerly upwelling IV (Fig. 2).
While we found the Peruvian upwelling and the adjacent waters to be only a moderate source
region for bromocarbons, iodocarbons were observed in high concentration of 10.9 (0.4 –
58.1) for $CH_2ClI$, 9.8 (1.1 – 35.4) for $CH_3I$ and 7.7 (0.2 – 32.4) pmol $L^{-1}$ for $CH_2I_2$ (Table 1,
Fig. 3a). These concentrations identify the Peruvian upwelling as a significant source region
of iodocarbons, especially considering the very short lifetimes of $CH_2I_2$ (10 min) and $CH_2ClI$
(9 h) in tropical surface water (Jones and Carpenter, 2005). Hot spots were upwelling regions
III and even more the less fresh upwelling of region IV (Fig. 2 and 3).
The occurrence of $CH_3I$ in the tropical oceans (up to 36.5 pmol $L^{-1}$) has previously been
attributed to a predominantly photochemical source (Richter and Wallace, 2004; Jones et al.,
2010), explaining its global hot spots in the subtropical gyres and close to the tropical western
boundaries of the continents (Ziska et al., 2013; Stemmler et al., 2014). Previous
measurements in the East Pacific obtained concentrations of up to 21.7 and of up to 8.8 pmol
$L^{-1}$ (Butler et al., 2007), but not directly in the upwelling.
No oceanic observations of $CH_2ClI$ and $CH_2I_2$ have been published so far for the tropical East
Pacific. Concentrations of $CH_2ClI$ of up to 24.5 pmol $L^{-1}$ were measured in the tropical and
subtropical Atlantic ocean (Abrahamsson et al., 2004; Chuck et al., 2005; Jones et al., 2010)
and up to 17.1 pmol $L^{-1}$ for $CH_2I_2$ (Jones et al., 2010; Hepach et al., 2015), which is lower but
in the range of our measurements from the Peruvian upwelling.




Correlations between the compounds indicate similar sources for all measured halocarbons,
except for $CH_2Br_2$, with upwelling region IV as hot spot area (Fig. 2). The strongest
correlation was found for $CH_3I$ with $CH_2ClI$ ($r_s = 0.83$). $CH_2I_2$ and $CH_2ClI$ are often found to
correlate very well with each other (Tokarczyk and Moore, 1994; Moore et al., 1996; Archer
et al., 2007), mostly attributed to the formation of $CH_2ClI$ during photolysis of $CH_2I_2$. In
comparison, the weaker correlation between $CH_2ClI$ and $CH_2I_2$ ($r_s = 0.59$) during our cruise
may be the result of additional sources for $CH_2ClI$ (see also section 5).
**4.2   Halocarbon distribution in depth profiles**
Depth profiles of halocarbons reveal maxima at the surface and around the Chl *a* maximum,
usually attributed to biological production of these compounds. $CHBr_3$ and $CH_2Br_2$ profiles
(not shown) showed distinct maxima in the deeper Chl *a* maximum during large part of the
cruise, while some profiles were characterized by elevated concentrations in the surface
usually associated with upwelling water. Both kinds of profiles are consistent with previous
studies finding maxima in the deeper water column in the open ocean and surface maxima in
upwelling regions (Yamamoto et al., 2001; Quack et al., 2004; Hepach et al., 2015). During
the northern part of M91 (upwelling III), $CH_2Br_2$ in the water column was more elevated than
$CHBr_3$, while during the remaining part of the cruise, $CHBr_3$ was usually higher.
Though most of the stations were characterized by subsurface maxima of iodocarbons, which
were mostly located between 10 and 50 m (see example in Fig. 4, upper panel), surface
maxima were often observed in upwelling region IV (see example in Fig. 4, lower panel), the
region with highest iodocarbon concentrations. Profiles with surface maxima were generally
characterized by much higher concentrations of these compounds. Similarly to very low
$CH_2I_2$ concentrations in the surface in the northern part of the measurements, it was also
hardly detected in the deeper water in this region (Fig. 4, upper panel). Surface maxima in
depth profiles of $CH_3I$ and $CH_2ClI$ were connected to surface maxima of several
phytoplankton species, mainly diatoms ($r_s = 0.57$ and $0.62$). Direct and indirect biological and
photochemical formation account as possible sources for these maxima. $CH_2I_2$ was usually
strongly depleted in the surface in contrast to the deeper layers due to its rapid photolysis,
which may also have been a source for surface $CH_2ClI$. Subsurface maxima occurred both
below and within the mixed layer (see the example in Fig. 4d indicated by the temperature-,
salinity- and density profiles). Maxima in the mixed layer probably appear because of very
fast production (Hepach et al., 2015), while maxima below the mixed layer are supported by
accumulation due to reduced mixing.





All five halocarbons were strongly depleted in waters below 50 m. These deeper layers were
also characterized by very low oxygen values, known as strong OMZ below the biologically
active layers (Karstensen et al., 2008). A possible reason for the strong depletion of the
halocarbons is their bacterial mediated reductive dehalogenation occurring under anaerobic
conditions (Bouwer et al., 1981; Tanhua et al., 1996).

## 5    Relationship of surface halocarbons to environmental parameters

Physical and chemical parameters as well as biological proxies such as TChl $a$ and
phytoplankton group composition were investigated using correlation analysis in order to
investigate marine sources of halocarbons.

### 5.1   Potential bromocarbon sources

Bromocarbons were weakly, but significantly anticorrelated with SSS and SST ($r_s$ between -
0.29 and -0.57), indicating sources in the upwelled water (Table 2). They showed a positive
correlation with diatoms ($r_s = 0.58$ for both compounds), the dominant phytoplankton group in
the region. Diatoms have already been found to be involved in bromocarbon production in
several laboratory and field studies (Tokarczyk and Moore, 1994; Moore et al., 1996; Quack
et al., 2007b; Hughes et al., 2013). Thus, these findings are in agreement with current
assumptions that diatoms may contribute directly or indirectly to bromocarbon production.
During M91, $CH_2Br_2$ was more abundant in cooler, nutrient-rich water than $CHBr_3$, leading to
a stronger correlation with TChl $a$ and SST, indicating an additional source associated with
fresh upwelling. No significant correlations were found for bromocarbons with
polysaccharidic DOM (Table 3), implying that DOM components analyzed during the cruise
were not involved in bromocarbon production, at least not in the upper water column.

### 5.2   Iodinated compounds and phyotplankton

In general, the iodocarbons correlated stronger with biological parameters than the
bromocarbons. Diatoms were found to correlate very strongly with all three iodocarbons ($r_s =$
0.73 with $CH_3I$, $r_s = 0.79$ with $CH_2ClI$ and $r_s = 0.72$). Weak but significant anticorrelations
with DIN:DIP and SST suggest that iodocarbons were associated with cool and slightly DIN
depleted water. The occurrence of large amounts of iodocarbons seemed to be associated with
an established diatom bloom. The production of $CH_3I$, $CH_2ClI$ and $CH_2I_2$ by a number of
diatom species has been observed in several studies before (Moore et al., 1996; Manley and





de la Cuesta, 1997), consistent with our findings. The very high correlation of cryptophytes
with iodocarbons was likely based on the co-occurrence of these species with diatoms (Table
2 and description in section 3).

## 5.3  Iodinated compounds and DOM

Correlations of the three iodinated compounds to polysaccharidic DOM components in
subsurface water revealed a strong relationship of the iodocarbon abundance with
polysaccharides and in particular uronic acids (Table 3). $CH_3I$ and $CH_2ClI$ showed strong
correlations with particulate uronic acids (both $r_s = 0.84$), total uronic acids ($r_s = 0.83$ and
0.88) and dissolved polysaccharides ($r_s = 0.82$ and 0.90). The correlations of $CH_2I_2$ with
polysaccharides were less strong, but significant ($r_s = 0.68$ with particulate, $r_s = 0.66$ with total
and $r_s = 0.55$ with dissolved). The above listed DOM components were also significantly
correlated to diatoms ($r_s = 0.68$ with polysaccharides and $r_s = 0.75$ with uronic acids), which
were a potential source for the accumulated organic matter in the subsurface. The exact
composition of surface water DOM is determined by the phytoplankton species producing the
DOM. Polysaccharides with uronic acids as an important constituent have for example been
shown to contribute largely to the DOM pool in a diatom rich region (Engel et al., 2012).
Hill and Manley (2009) tested several diatom species for their production of halocarbons in a
laboratory study, and suggested that a major formation pathway for polyhalogenated
compounds may actually not be from direct algal production, but rather indirectly through
their release of hypoiodous (HOI) and hypobromous acid (HOBr), which then react with the
present DOM (Liu et al., 2015). The formation of HOI and HOBr within the algae is
enzymatic with possible chloroperoxidase (CPO), BPO and iodoperoxidase (IPO)
involvement. While CPO and BPO may produce both HOBr and HOI, IPO only leads to HOI.
Moore et al. (1996) suggested that the occurrence of BPO and IPO in the phytoplankton cells
may be highly species dependent. This leads to the assumption that diatoms abundant in the
Peruvian upwelling contained more IPO than BPO, which could explain the higher abundance
of iodocarbons relative to bromocarbons during M91.
The formation of $CH_3I$ through DOM may be different than the production of $CH_2ClI$ and
$CH_2I_2$. While $CH_2I_2$ is suggested to be formed via haloform-type reactions (Carpenter et al.,
2005), $CH_3I$ is produced using a methyl-radical source (White, 1982). The relationship of
$CH_3I$ with DOM can be the result of both photochemical and biological production pathways:
DOM, which was observed in high concentrations in the biologically productive waters, can





act as the methyl-radical source during photochemical production of $CH_3I$ (Bell et al., 2002).
A second possible biological pathway of methyl iodide production takes place via bacteria
and micro algae, which can utilize methyl transferases in their cells. HOI plays a significant
role in this production pathway by providing the iodine to the methyl group (Yokouchi et al.,

5   2014).

In conclusion, the Peruvian upwelling was a strong source for the iodocarbons $CH_3I$, $CH_2ClI$
and $CH_2I_2$, and a weaker source for the bromocarbons $CHBr_3$ and $CH_2Br_2$. We propose a
formation mechanism for this region as described in Fig. 5 based on measurements of short-
lived halocarbons and biological parameters during M91. Diatoms, which can contain the
necessary enzymes for halocarbon formation, were identified as important source based on
their strong correlations with the bromo- and iodocarbons and with polysaccharidic DOM.
The very good correlations of iodocarbons with polysaccharides and uronic acids are a hint
that these DOM components may have been important substrates for iodocarbon production
potentially produced from the present diatoms. The higher iodocarbon concentrations can
likely be explained by phytoplankton species containing more IPO than BPO, leading to a
stronger production of iodocarbons. Additionally, the particular type of DOM may also have
regulated the production of specific halocarbons (Liu et al., 2015), in this case $CH_3I$, $CH_2ClI$
and $CH_2I_2$.
One interesting feature of our analysis is the fact that $CH_2I_2$ when compared to the other two
iodocarbons showed weaker correlations with the polysaccharides possibly due to its shorter
surface water lifetime. Moreover, $CH_2I_2$ and $CH_2ClI$ showed weaker correlations in the
Peruvian upwelling than during other cruises in the tropical Atlantic, namely MSM18/3
(Hepach et al., 2015) and DRIVE (Hepach et al., 2014). Combining the two arguments of a
short $CH_2I_2$ lifetime and only a weak correlation between $CH_2I_2$ and $CH_2ClI$, this may
indicate an additional source for $CH_2ClI$ similar to $CH_3I$, explaining why $CH_3I$ and $CH_2ClI$
correlate much better with each other than with $CH_2I_2$.
**6    From the ocean to the atmosphere**
**6.1    Sea-to-air fluxes of iodocarbons**
Due to high oceanic iodocarbon concentrations measured in sea surface water of the Peruvian
upwelling and despite the moderate prevailing wind speeds of 6.17 (0.42 – 15.47) m s$^{-1}$, high
iodocarbon sea-to-air fluxes were calculated in contrast to the rather low bromocarbon





emissions during this M91 cruise (Fuhlbrügge et al., 2015a). The highest average fluxes of the
three iodocarbons of 954 (21 − 4686) were calculated for $CH_3I$, followed by 834 (-24 − 5652)
for $CH_2ClI$, and finally 504 (-126 − 2546) pmol $m^{-2}$ $h^{-1}$ for $CH_2I_2$ (Table 1). These were on
average 4 to 7 times higher than $CHBr_3$ and 2 to 4 times higher than the $CH_2Br_2$ sea-to-air
fluxes during the cruise.
Our estimated fluxes of $CH_3I$ are in the range of emissions calculated for the tropical and
subtropical Atlantic of 625 to 2154 pmol $m^{-2}$ $h^{-1}$ (Chuck et al., 2005; Jones et al., 2010).
Moore and Groszko (1999), who performed a study between 40° N and 40° S close to our
investigation region but not covering the Peruvian upwelling, calculated on average 666 pmol
$m^{-2}$ $h^{-1}$, which is 0.7 times our flux. Sea-to-air fluxes of $CH_2ClI$ from the same studies were
reported to range on average between 250 and 1138 pmol $m^{-2}$ $h^{-1}$ with the largest fluxes
originating from the Mauritanian upwelling region. These are 0.3 to 1.4 times the fluxes we
calculated, showing that the Peruvian upwelling region is at the top end of oceanic $CH_2ClI$
emissions. We are only aware of two studies focusing on emissions of $CH_2I_2$ from the
Atlantic tropical ocean (Jones et al., 2010; Hepach et al., 2015) which are on average 0.2,
respectively 1.4 times the fluxes from the tropical East Pacific. The larger sea-to-air fluxes
reported in Hepach et al. (2015) from the equatorial Atlantic cold tongue are mainly a result
of much lower atmospheric mixing ratios there, increasing the concentration gradient, and
additionally higher wind speeds, increasing the exchange coefficient $k_w$.
Summarizing this section, the large production of iodocarbons in the Peruvian upwelling led
to enhanced emissions of these compounds to the troposphere despite very low wind speeds.
An additional factor influencing halocarbon emissions is the low height and insolation of the
MABL, where halocarbons accumulate above the air-sea interface. The large sea-to-air fluxes
and low wind speeds should result in high tropospheric iodocarbons, which was indeed
observed and is discussed in the following section.
**6.2    Atmospheric iodocarbons**
Atmospheric mixing ratios of the three iodocarbons were elevated during M91 with up to 3.2
ppt for $CH_3I$, up to 2.5 ppt for $CH_2ClI$ and up to 3.3 ppt for $CH_2I_2$ (Table 1), likely a result of
the strong production and emissions of these compounds.
$CH_3I$ data were generally elevated in comparison to other eastern Pacific measurements of up
to 2.1 ppt $CH_3I$ (Butler et al., 2007), but lower than in the tropical central and East Atlantic
around the equator, characterized by higher atmospheric $CH_3I$ of over 5 ppt (Ziska et al.,





2013). Both $CH_2ClI$ and $CH_2I_2$ were also elevated in comparison to previous oceanic
measurements, where e.g. 0.01 to 0.99 ppt $CH_2ClI$ were measured for remote locations in the
Atlantic and Pacific (Chuck et al., 2005; Varner et al., 2008) and only up to 0.07 ppt were
reported for $CH_2I_2$ at a remote site in the Pacific (Yokouchi et al., 2011). Coastal areas with
high macroalgal abundance were characterized by high $CH_2ClI$ of up to 3.4 ppt (Varner et al.,
2008) and up to 3.1 ppt $CH_2I_2$ (Carpenter et al., 1999; Peters et al., 2005), while 19.8 ppt
(Peters et al., 2005) were measured at Mace Head, Ireland and Lilia, Brittany in the North
Atlantic.
The different atmospheric lifetimes of the three iodocarbons, ranging between 4 d ($CH_3I$), 9 h
($CH_2ClI$) and 10 min ($CH_2I_2$) (Carpenter and Reimann, 2014), partly explain the observed
differences in their distributions. Although atmospheric $CH_3I$ was generally elevated in
regions of high oceanic $CH_3I$ (Fig. 3) in upwelling regions III and IV, the atmospheric and
oceanic data did not show a significant correlation. The $CH_3I$ lifetime of several days allows
atmospheric $CH_3I$ to mix within the MABL, possibly masking a correlation between local
source regions and elevated mixing ratios.
The two shorter-lived iodinated compounds $CH_2ClI$ and $CH_2I_2$ generally showed a stronger
influence of local marine sources. Both species correlate significantly with their oceanic
concentrations with $r_s = 0.60$ ($CH_2ClI$) and $r_s = 0.64$ ($CH_2I_2$). Oceanic $CH_2ClI$ and $CH_2I_2$ were
emitted into the boundary layer where they could accumulate during night (see comparison
with global radiation in Fig. 3b), and were rapidly degraded during day time via photolysis,
which is their main sink in the troposphere (Carpenter and Reimann, 2014). Moderate average
wind speeds in the upwelling regions (Fig. 6) and stable atmospheric boundary layer
conditions (Fuhlbrügge et al., 2015a) supported the accumulation of these compounds.
The Peruvian upwelling was in general characterized by elevated atmospheric iodocarbons as
a result of their large sea-to-air fluxes caused by strong biological production. The upwelling
could sustain elevated atmospheric levels of e.g. $CH_3I$, could trap iodocarbons and their
degradation products in a stable MABL, and may have therefore contributed significantly to
the tropospheric inorganic iodine budget, which is discussed in the following.
**6.3   Contributions to tropospheric iodine**
After their emission from the ocean and their chemical degradation in the marine boundary
layer, iodocarbons contribute to the atmospheric inorganic iodine budget, $I_y$. The importance
of this contribution compared to abiotic sources is currently under debate and analyzed for



various oceanic environments (Mahajan et al., 2010; Grossmann et al., 2013; Prados-Roman
et al., 2015). So far, no correlations of IO with the organic iodine precursor species have been
observed (Grossmann et al., 2013) and correlations between IO and Chl *a* were often found to
be negative (Mahajan et al., 2012; Gómez Martín et al., 2013). Chemical modelling studies,
undertaken to explain the contributions of organic and inorganic oceanic iodine sources,
simulated that only a small fraction of the atmospheric IO stems from the organic precursors
with estimates of about 25 % on a global average (Prados-Roman et al., 2015). Both
arguments, the missing correlations and the small contributions, indicate that the organic
source gas emissions play a minor role for the atmospheric iodine budget. Given the special
conditions of the Peruvian upwelling with cold nutrient rich waters, the strong iodocarbon
sources and a stable MABL and trade inversion, it is of interest to analyze the local
contributions to the atmosphere in this region and to compare with estimates from other
oceanic environments.
We focus our analysis on the section of the cruise where MAX-DOAS measurements of IO
and simultaneous iodocarbon measurements in the surface water and atmosphere were made
(roughly south of 10° S). Tropospheric VCDs of IO in the range of $2.5 - 6.0 \times 10^{12}$ molec cm$^{-2}$
were inferred from the MAX-DOAS measurements. Similar VCDs of IO were reported by
Schönhardt et al. (2008) based on remote satellite measurements from SCIAMACHY.
Volume mixing ratios of IO along the cruise track (Fig. 7a) derived from the MAX-DOAS
measurements show a pronounced variability and maxima close to upwelling regions II and
IV. Daytime averaged IO volume mixing ratios are displayed in Fig. 7b and range between
0.8 ppt (on December 12 and December 22) and 1.5 ppt (on December 26). Overall the
daytime IO abundance in the MABL above the Peruvian upwelling was relatively high
compared to measurements from the nearby Galapagos islands (~ 0.4 ppt) (Gómez Martín et
al., 2013) and  from other tropical oceans such as the Malaspina 2010 circumnavigation (0.4 –
1 ppt) (Prados-Roman et al., 2015). Other measurement campaigns such as the Cape Verde
measurements (Read et al., 2008) or the TransBrom Sonne in the West Pacific (Grossmann et
al., 2013) found similar IO mixing ratios with values above 1 ppt, but significantly lower IO
VCDs in case of the latter.
The M91 cruise track crisscrossed the waters between the coast and 200 km offshore multiple
times, providing a comprehensive set of measurements over a confined area (see Fig. 7a) and
allowing us to analyze the relation between IO and organic precursors. Assuming constant
emissions over the cruise period we can link the oceanic sources with atmospheric IO
observations at locations reached after hours to days of atmospheric transport. Therefore, we



released FLEXPART trajectories from all sea surface measurement locations continuously
over the whole measurement time period from December 8 to December 26 loaded with the
oceanic organic iodine as prescribed by the observed iodocarbon emissions. Based on the
simulations of transport and chemical decay described in Section 2.5, we derived organic and
inorganic iodine mixing ratios individually for each air parcel. Mixing with air parcels
impacted by other source regions was not taken into account. FLEXPART-based IO
originating from organic precursors was derived as mean values over all air parcels in the
MABL coinciding with the MAX-DOAS measurement locations within an area of 5 km x 5
km. Simplifying assumptions of a prescribed inorganic iodine lifetime (2 days) and IO to $I_y$
ratio (0.15 to 0.3) were made to derive the IO mixing ratios. Uncertainties were estimated
based on additional runs with varying atmospheric lifetime of inorganic iodine (1 − 3 days).
For the first part of the cruise from December 8 to December 18, FLEXPART-derived IO
mixing ratio estimates at the MAX-DOAS measurement locations (red line in Fig. 7b and c)
explain between 40 and 70 % (55 % on average) of the measured IO assuming a  lifetime for
inorganic iodine of 2 days. As a consequence, about 0.5 ppt of IO is expected to originate
from other, likely inorganic, iodine sources. For the scenario of a shorter $I_y$ lifetime (one day),
we find that the organic sources explain about 30 % of the IO and for a relatively long
lifetime (three days), 80 % can be explained. In general, the air masses were transported along
the coast in northwest direction and organic sources contribute to the IO budget along this
transport path. Most of the IO results from $CH_2ClI$ (Fig. 7c) which was transported some
hours northwestwards (lifetime of 9 hours) before contributing to the atmospheric inorganic
iodine budget.
For the second part of the cruise from December 19 to December 26, the amount of IO
estimated from organic precursors was much smaller and often close to zero. This very small
organic contribution was caused by two facts. First, the instantaneous sources during the last
part of the cruise were much smaller (Fig. 6b − d) and second, further southward situated
sources, which also influence the iodine abundance in the cruise track region were not
analyzed and thus not included in the simulations. Because of missing information on the
source strength and distribution southwards of the cruise track, a proper comparison is only
possible for the first part of the cruise before December 19. However, given that the MAX-
DOAS measurements of IO remained relatively high during the second part of the cruise, it is
likely that additional significant organic iodine sources existed further southwards. This
assumption is also supported by the fact that atmospheric $CH_3I$ mixing ratios remained
relatively high during the second part of the cruise (50 % compared to the earlier part, see Fig.



3b) while the water concentrations were close to zero. Consequently, a source region of $CH_3I$
must have existed further southwards contributing to the observed mixing ratios of $CH_3I$ and
IO after some hours to days of atmospheric transport.
While the contribution of organic iodine to IO during the first part of the cruise is
considerably higher than found in other regions, the amount of inorganic iodine precursors of
0.5 ppt necessary to explain total IO is very similar to the one derived in other studies
(Prados-Roman et al., 2015). The higher organic contribution was consistent with the fact that
there was an overall higher IO abundance compared to most other campaigns. Instantaneous
IO and organic source gas emissions during M91 were not directly correlated. However,
taking the transport within the first hours and days into account enables us to explain a
considerable part of the atmospheric IO variations with the variability of the oceanic organic
sources (Fig. 7b). Overall, we conclude that for the Peruvian upwelling region with special
conditions in the ocean and atmosphere, higher iodocarbon sources lead to larger IO
abundances while the absolute inorganic contribution is similar to other regions.

## 16   7   Conclusions

The Peruvian upwelling at the west coast of South America was characterized for halocarbons
for the first time during the M91 cruise. We measured moderate concentrations of the
bromocarbons $CHBr_3$ and $CH_2Br_2$, while we observed exceptionally high concentrations of
the iodocarbons $CH_3I$, $CH_2ClI$ and $CH_2I_2$ in the surface seawater.
$CHBr_3$ and $CH_2Br_2$ were significantly correlated with TChl *a* and diatoms, suggesting
biological formation of these compounds. Higher correlations of diatoms were found with the
three iodocarbons, and even stronger correlations of the iodocarbons with the DOM
components polysaccharides and uronic acids were observed. The polyhalogenated
compounds $CH_2ClI$ and $CH_2I_2$ were potentially formed via these DOM components with the
likely involvement of diatoms. $CH_3I$ may have been formed via photochemistry from the
large pool of observed DOM and/or biologically via methyl transferases in micro algae and
bacteria. The production of iodocarbons from DOM via the proposed mechanisms seems to
have exceeded the bromocarbon production in the region in contrast to several previous
studies in tropical Atlantic upwelling regions (Hepach et al., 2014, 2015).
Depth profiles showed subsurface maxima, common in the open ocean, and very pronounced
and elevated surface maxima in regions of highest underway iodocarbon concentrations. The



surface water was always depleted in $CH_2I_2$ with respect to the underlying water column due
to its very rapid photolysis. The OMZ at depth was strongly depleted in all five measured
halocarbons, suggesting an effective sink in the oxygen depleted waters.
The high oceanic iodocarbon concentrations and elevated emissions also led to elevated
atmospheric mixing ratios in the marine boundary layer. Atmospheric $CH_2ClI$ and $CH_2I_2$
showed clear diurnal cycles, accumulating during night and decreasing rapidly during day
time. Despite previous suggestions that the tropospheric iodine loading is mainly a product
from direct emission of HOI and $I_2$, we calculated important contributions of iodocarbons to
the observed IO levels. Using FLEXPART, we estimated a contribution of combined
iodocarbon fluxes to IO of 30 to 80 % assuming an inorganic iodine lifetime between 1 and 3
days. This contribution of organoiodine is much higher than previously assumed (Prados-
Roman et al., 2015), suggesting that iodocarbons therefore may contribute significantly to
tropospheric iodine levels in regions of strong iodocarbon production mediated by
phytoplankton (diatoms) and bacteria.
Our observations reveal several uncertainties which need to be addressed in the future to
better constrain the halocarbon budget and understand its role in a changing climate. Further
studies of upwelling regions need to be performed in different seasons and years, since these
regions are impacted by synoptic and climatic conditions, which are expected to have an
impact on the strength of halocarbon emissions. A regular monitoring and better knowledge
of halocarbon sources and emissions is severely needed, since these have numerous
implications for atmospheric processes such as ozone chemistry and aerosol formation, which
have been investigated in several atmospheric modeling studies. These studies have mostly
applied Chl $a$ as proxy for halocarbon emissions. However, the potential involvement of
DOM in the production of both iodo- and bromocarbons and the often weak correlation to Chl
$a$ in the field raises the question whether Chl $a$ is a suitable parameter to estimate halocarbon
concentrations. Laboratory studies are therefore crucial to help identifying more adequate
parameters for predicting halocarbons in the ocean. Associated with the involvement of DOM
in iodocarbon production is the occurrence of the relevant DOM components in large
concentrations in the sea surface microlayer (SML) (Engel and Galgani, 2015). The SML has
been shown to cover a wide range of oceanic regions (Wurl et al., 2011), which could
represent a significant additional source to atmospheric iodocarbons. The potential of the
SML to produce $CH_2I_2$ has been previously suggested by Martino et al. (2009), who proposed
that HOI converted from iodide in the SML may react with the present DOM, which may
apply to $CH_2ClI$ and $CH_3I$ as well. Furthermore, the SML is in direct contact to the





atmosphere, and the direct exposure to light may enhance halocarbon emissions (see also Fig.
5). The influence of these halocarbon emissions on the tropospheric halogen loading is still
very much under debate, and our results underline the importance to constrain the actual
contribution of these compounds to tropospheric halogen chemistry.
**Acknowledgements**
We thank the chief scientist of the cruise M91 Hermann Bange, as well as the captain and the
crew of the RV *Meteor* for their support. We would like to acknowledge Sonja Wiegmann for
pigment analysis, Kerstin Nachtigall for nutrient measurements, and Stefan Raimund and
Sebastian Flöter for helping with halocarbon measurements. This work was part of the
German research projects SOPRAN II (grant no. FKZ 03F0611A) and III (grant no. FKZ
03F0662A) funded by the Bundesministerium für Bildung und Forschung (BMBF). Astrid
Bracher's contribution was funded by the Total Foundation project „Phytoscope".



**Tables**
Table 1. Environmental parameters, as well as halocarbons in water, air and sea-to-air fluxes
during the cruise. Means of sea surface temperature (SST), sea surface salinity (SSS) and
wind speed are for 10-min-averages.

| Parameter | | Unit | Mean (min - max) |
|---|---|---|---|
| SST | | °C | 19.4 (15.0 - 22.4) |
| SSS | | | 34.95 (34.10 - 35.50) |
| TChl $a$ | | µg L$^{-1}$ | 1.80 (0.06 - 12.65) |
| Wind speed | | m s$^{-1}$ | 6.17 (0.42 - 15.47) |
| CHBr$_3$ | Water | pmol L$^{-1}$ | 6.6 (0.2 - 21.5) |
| | Air | ppt | 2.9 (1.5 - 5.9) |
| | Sea-to-air flux | pmol m$^{-2}$ h$^{-1}$ | 130 (-550 - 2201) |
| CH$_2$Br$_2$ | Water | pmol L$^{-1}$ | 4.3 (0.2 - 12.7) |
| | Air | ppt | 1.3 (0.8 - 2.0) |
| | Sea-to-air flux | pmol m$^{-2}$ h$^{-1}$ | 273 (-128 - 1321) |
| CH$_3$I | Water | pmol L$^{-1}$ | 9.8 (1.1 - 35.4) |
| | Air | ppt | 1.5 (0.6 - 3.2) |
| | Sea-to-air flux | pmol m$^{-2}$ h$^{-1}$ | 954 (21 - 4686) |
| CH$_2$ClI | Water | pmol L$^{-1}$ | 10.9 (0.4 - 58.1) |




| | | | |
|---|---|---|---|
| | Air | ppt | 0.4 (0 - 2.5) |
| | Sea-to-air flux | pmol m$^{-2}$ h$^{-1}$ | 834 (-28 - 5652) |
| | Water | pmol L$^{-1}$ | 7.7 (0.2 - 32.4) |
| $CH_2I_2$ | Air | ppt | 0.2 (0 - 3.3) |
| | Sea-to-air flux | pmol m$^{-2}$ h$^{-1}$ | 504 (-126 - 2546) |



1  Table 2. Spearman's rank correlation coefficients of correlations of halocarbon with several ambient parameters, as well as biological proxies.
2  Bold numbers indicate correlations that are significant with $p < 0.05$ with a sample number of 107 for all environmental data and 46 for all
3  phytoplankton and nutrient data considering all collocated surface data.

|  | CHBr$_3$ | CH$_2$Br$_2$ | CH$_3$I | CH$_2$ClI | CH$_2$I$_2$ | SST | SSS | Global radiation | Diatoms | Crypto-phytes | Dino-flagellates | TChl $a$ |
|---|---|---|---|---|---|---|---|---|---|---|---|---|
| DIN:DIP | -0.26 | -0.34 | **-0.38** | **-0.30** | -0.32 | 0.00 | **0.41** | 0.13 | -0.18 | -0.17 | -0.10 | -0.08 |
| TChl $a$ | **0.48** | **0.56** | **0.73** | **0.74** | **0.70** | **-0.82** | **-0.77** | -0.20 | **0.93** | **0.85** | **0.33** | |
| Dino-flagellates | 0.15 | 0.28 | 0.15 | 0.17 | 0.21 | -0.23 | -0.22 | -0.01 | **0.38** | 0.26 | | |
| Crypto-phytes | **0.38** | **0.54** | **0.61** | **0.61** | **0.64** | **-0.74** | **-0.79** | -0.18 | **0.73** | | | |
| Diatoms | **0.58** | **0.58** | **0.73** | **0.79** | **0.72** | **-0.76** | **-0.72** | -0.18 | | | | |
| Global radiation | 0.14 | -0.10 | -0.22 | -0.03 | -0.08 | **0.20** | 0.12 | | | | | |
| SSS | **-0.44** | **-0.48** | **-0.75** | **-0.69** | **-0.45** | **0.68** | | | | | | |
| SST | **-0.29** | **-0.57** | **-0.52** | **-0.62** | **-0.58** | | | | | | | |
| CH$_2$I$_2$ | **0.60** | 0.43 | **0.66** | **0.59** | | | | | | | | |
| CH$_2$ClI | **0.64** | **0.70** | **0.83** | | | | | | | | | |
| CH$_3$I | **0.66** | **0.46** | | | | | | | | | | |

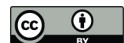



$CH_2Br_2$    **0.56**





1  Table 3. Correlations of halocarbons with combined high molecular weight (HMW)

2  carbohydrates (CCHO) and uronic acids (URA) from subsurface samples (T – total, d –

3  dissolved, P – particulate) with a sample number of 29 for each variable.

|  | $CHBr_3$ | $CH_2Br_2$ | $CH_3I$ | $CH_2ClI$ | $CH_2I_2$ |
|---|---|---|---|---|---|
| TCCHO | 0.15 | 0.28 | **0.78** | **0.82** | **0.66** |
| dCCHO | 0.39 | 0.48 | **0.82** | **0.90** | **0.55** |
| PCCHO | -0.06 | -0.10 | **0.61** | **0.64** | **0.68** |
| TURA | 0.31 | 0.34 | **0.83** | **0.88** | **0.52** |
| dURA | -0.18 | 0.42 | **0.48** | **0.79** | **0.50** |
| PURA | 0.37 | 0.22 | **0.84** | **0.84** | **0.54** |





# 1  Figures

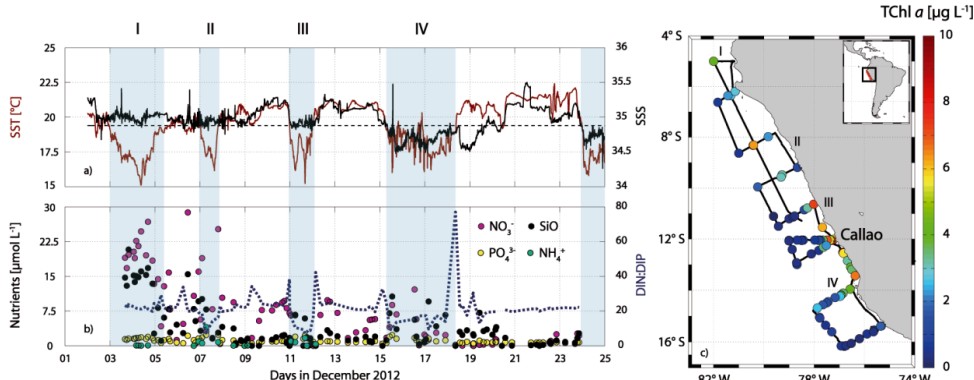

Fig. 1. Ambient parameters during the M91 cruise: SST (dark red) and sea surface salinity (SSS) (black) in a) with the dashed line as the mean SST. Nutrients (purple is nitrate – $NO_3^-$, yellow is phosphate – $PO_4^{3-}$, black is silicate – $SiO_2$, light cyan is ammonium – $NH_4^+$) with the N to P ratio (dark blue dashed line) are shown in b). Total chlorophyll $a$ (TChl $a$) is shown in the map in c). The light blue shaded areas stand for the regions where SST is below the mean, indicating upwelling of cold water.



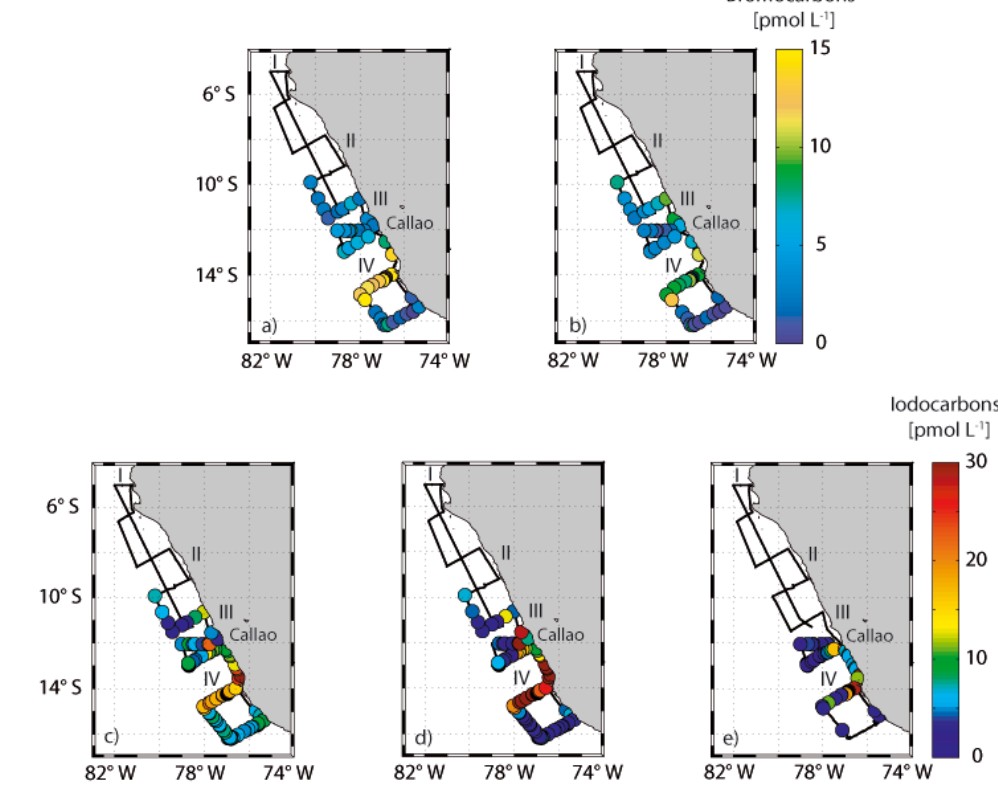



Fig. 2. Halocarbon surface water measurements are shown in a) and b) for the bromocarbons
(note the colorbar oin the upper panel) with a – $CHBr_3$ and b – $CH_2Br_2$. Iodocarbons can be
found in c) – e) (note the colorbar in the lower panel) with c – $CH_3I$, d – $CH_2ClI$ and e –
$CH_2I_2$.

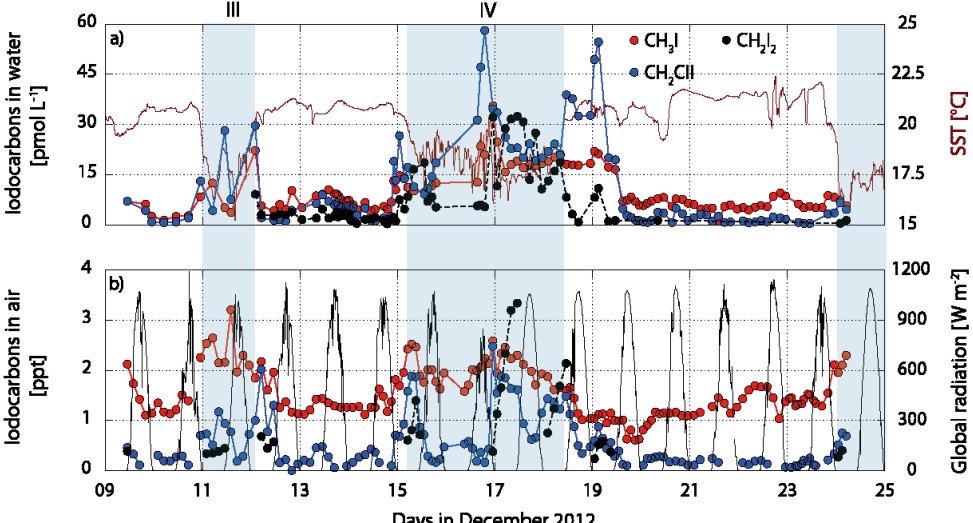

Fig. 3. Surface water measurements of iodocarbons are presented in a) with $CH_3I$ in red,
$CH_2ClI$ in blue and $CH_2I_2$ in black on the left side along with SST (dark red) on the right side.
Additionally, atmospheric mixing ratios of $CH_3I$ (red), $CH_2ClI$ (blue) and $CH_2I_2$ (grey) on the
left side together with global radiation (black) on the right side are depicted in b). Note that all
times are in UTC.



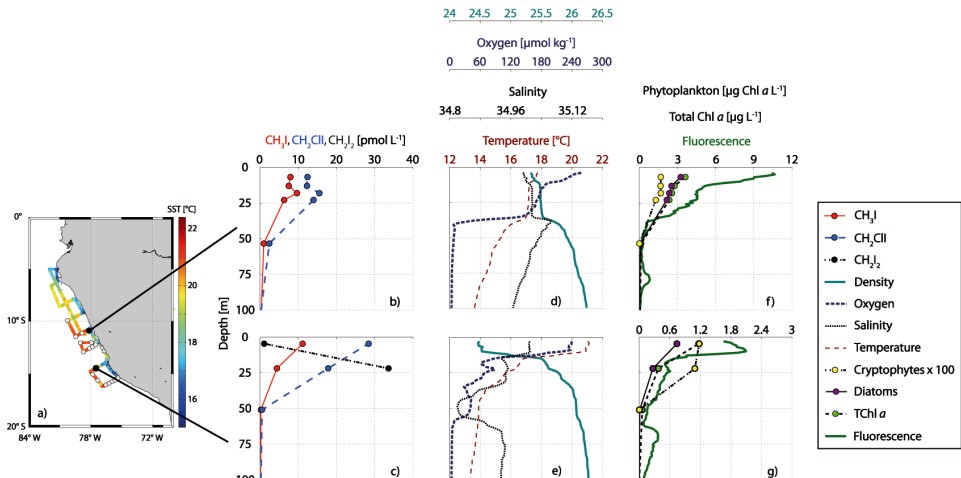

Fig. 4. A cruise map including all CTD stations and SST is shown in a), while selected depth
profiles of iodocarbons can be seen in a) – b), together with ambient parameters such as
potential density (cyan), oxygen (dark blue), salinity (black) and temperature (dark red) in d)
– e), as well as phytoplankton groups (cryptophytes and diatoms), total chlorophyll a and
fluorescence in f) – g). $CH_2I_2$ was undetectable at the first station (see also consistence with
surface data).



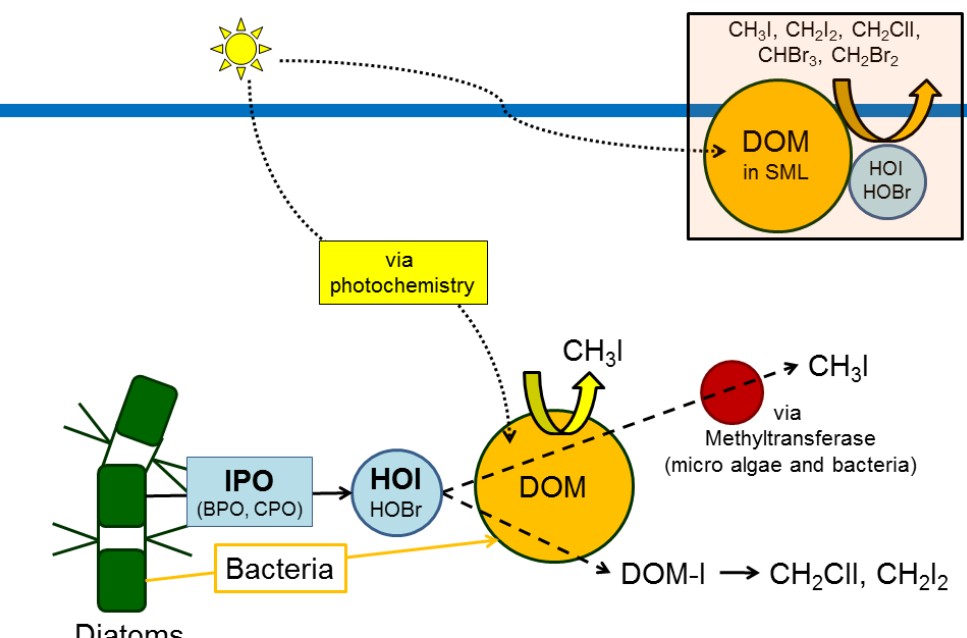

Fig. 5. Proposed mechanisms for formation of iodocarbons – release of HOI with the help
ofvia iodoperoxiases ( IPO) and reaction with DOM (dissolved organic matter ( DOM ) via
iodine binding to DOM (DOM-I) to form $CH_2ClI$ and $CH_2I_2$. $CH_3I$ formation via
photochemistry and/or biological formation via methyltransferases. The box indicates
potential formation of halocarbons from DOM in the sea surface microlayer SML.





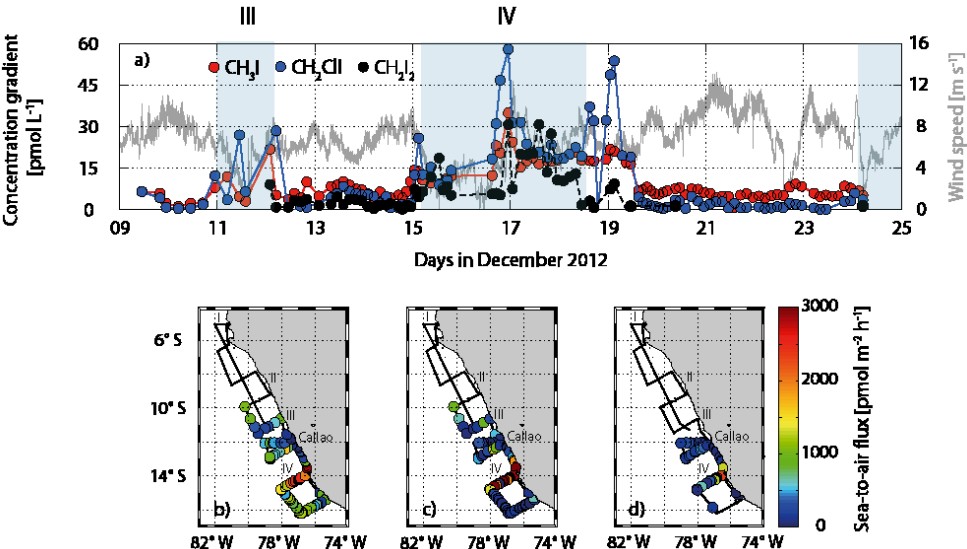

Fig. 6. The concentration gradient of $CH_3I$, $CH_2ClI$ and $CH_2I_2$ along with wind speed (grey) is shown in a), while the sea-to-air flux is depicted in b) for $CH_3I$, c) for $CH_2ClI$ and d) for $CH_2I_2$. Note the color bar on the right.

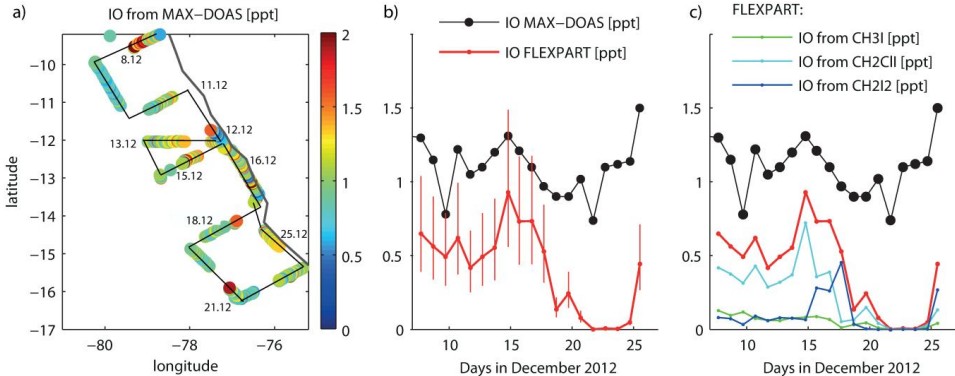

Fig. 7. MAX-DOAS measurements of IO during the M91 campaign along the cruise track are shown a). Daytime averaged IO values from MAX-DOAS and coincident FLEXPART values are provided in b). The vertical bars correspond to uncertainties associated with the inorganic iodine lifetime in the MABL (1 − 3 days) and the daytime fraction of IO to $I_y$ (0.15 to 0.3).



1    Contributions of the three oceanic iodocarbon sources to the modelled FLEXPART IO are

2    also given in c).



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
