# Peer review of "Biogenic halocarbons from the Peruvian upwelling region as tropospheric halogen source"

_Atmospheric Chemistry and Physics, 2016_

## Referee Comment (RC1) · Anonymous Referee #1 · 5 Apr 2016

General comments This manuscript by Hepach et al. reported their observations on a series of bromo- and iodocarbons from surface ocean as well as subsurface waters in the Peruvian upwelling zone. The authors further investigated into possible sources of the halocarbons and the contribution of organoiodine to stratosphere iodine loading. This manuscript is relatively well written and discussed drivers for the production of the halocarbons and transport of iodinated species, in addition to reporting data. I believe it is a new contribution to the scientific community. The authors also proposed directions for future studies for addressing the halocarbon budgets. I believe this manuscript is suitable for publication in ACP with minor revisions. Please see specific comments suggested below.

Specific comments

[Figure]

Page 2 Line 2: delete "halogenated short-chained hydrocarbons" for "halocarbons" is clear enough for this definition.

Page 2 Line 6: should read "Peruvian upwelling zone (or regions)".

Page 2 Line 23: perhaps use "very short-lived substances (VLS)" following the WMO terminology?

Page 3 Line 16: please clarify this sentence by adding "in seawater" after "for both compounds".

Page 3 Line 29: should read "the main sinks for both CH2I2 and CH2ClI are..."

Page 4 Line 14: replace "the latter two compounds" with "CH2ClI and CH2I2" to avoid confusion.

Page 5 Line 2: it is a bit confusing here, is it "every 3 hours"?

Page 5 Line 6 and 7: were purge efficiency measured for these gases?

Page 5 Line 11: please specified how the gas samples were stored – stainless steel canisters?

Page 5 Line 25: Cyanobacteria may pass through the GF/F filters at the initial filtering (i.e. before the filter pore size decreased as materials accumulated), which may affect the quantification of the cyanobacteria marker pigments. Did the authors estimate such a biomass lost?

Page 5 Lines 28 to Page 6 Lines 1 to 7: The DOM samples were collected from 20cm and the gases were collected from about 6 to 7 m, which were not exactly parallel samples. Some DOM can be recycled relatively fast. In addition, DOM at surface ocean may be degraded via photolysis. I suggest the authors to also report the mix layer depth and possible residence times for the DOM compounds they measured, such that a valid argument can be made about those DOM were well mixed within the mixed layer and hence the depth difference would not affect the data analysis and

interpretation.

Page 8 Lines 1 to 15: I suggest move this to before the FLEXPART model simulation.

Page 9 Line 2: change "correlated very well" to "significantly correlated".

Page 11 Line 11: I suggest the authors also include the depth profile of the bromocarbons.

Page 11 Lines 22 to 24: this sentence is a bit confusing, please rephrase.

Page 12 Line 22: Liu et al., 2015 tested a series of carbohydrates, and found that these DOM moieties were not fast reacting substrates for CHBr3, which seems to be consistent with findings in this study. In addition, bromocarbon formations via HOBr reaction are potentially DOM moiety specific. In Liu et al., 2013, no correlations were observed between the bromocarbons and total dissolved organic carbon.

Page 13 Line 12: the bulk DOM may correlate better with total biomass (estimated from TChla).

Page 13 Lines13 to 15: I suggest change "is determined by the phytoplankton species" to "is determined by ecosystem compositions", because DOM contribution is not governed by phytoplankton species alone.

Page 13 Line 21: Please also cite Lin and Manley 2012, who also tested bromocarbon formations using different molecular weight natural DOM as substrates.

Page 13 Lines 24 to 27: The authors depicted possible abiotic sources of HOI and HOBr in Fig 5. I would suggest the authors also put the abiotic sources of HOI and HOBr into this context (see Carpenter et al., 2005).

Page 14 Line 6: I suggest remove "In conclusion" here.

Page 31 Fig 5: The conception model figure is a bit confusing on the CH3I part via methyltransferase, for it is an intracellular enzyme. Thus the reaction is likely occurring

inside the cell. However, figure seems to depict an extracellular reaction.

---

## Referee Comment (RC2) · Anonymous Referee #2 · 11 Apr 2016

General Comments

This manuscript by Hepach and co-authors presents gas-phase and oceanic observations of halogenated VOCs, including CHBr3, CH2Br2, CH2ClI, CH3I, and CH2I2 from a ship cruise in the eastern tropical Pacific Ocean. In addition to the concentrations and sea-air flux calculations, the analysis includes correlations to phytoplankton groups measured in the surface water along the cruise path. The paper is reasonably well written and the many of the data presented are new observations. The paper should be published in ACP after addressing the following minor corrections.

Specific Comments

Page 5, line 2 – Three hourly is not explicit – every three hours, or three samples per hour? And was this day and night? What were the samples taken in? I would like to

see a little more detail, even though the system was described in another paper.

Page 5, line 7 – replace "lay" with "was".

Page 5, line 9 – "set up problems" sounds odd. Perhaps "instrument issues?"

Page 5, line 10 – 20 m "above sea level?" How were these samples stored? Stainless canisters? Glass flasks?

Page 5, line 11 – "starting on December 1..."

Page 5, line 23 – replace "build up" with "comprise"

Page 5, line 26 – "a" should be italicized.

Page 7, line 20 – The authors explain that they do not include detailed tropospheric iodine chemistry, and specify what they, explicit removal of HOI, HI, IONO2, and IxOy through scavenging or 20 heterogeneous recycling of HOI, IONO2, and INO2 on aerosols, and then reference Saiz-Lopez et al., 2014. It should be made clear whether or not Saiz-Lopez did or did not omit iodine removal . I.e., "we didn't do x, y and z (reference)" - is the reference an example where x, y and z were omitted, or not omitted?

Page 8 – section 2.6 and section 2.5 should be swapped (i.e., measurement methods before model description.)

Page 10, lines 1-4 – The suggestion that the observations from this work "compare well" with observations from Liu et al. (2013) needs to be backed up with something more quantitative. CHBr3 seems to have a similar range, but the CH2Br2 range from Liu et al. are about half the values from the current work. Can you be more specific about the region covered by Liu et al., i.e. where the observed maxima were located?

Page 11, line 11 – "... during a large part... "

Page 11, lines 18-21 – I have a bit of an issue with this interpretation. In Figure 4, we

see that there were four subsurface measurements made between the surface and a depth of 25 m, and one single subsurface maxima is shown at a depth of about 20 m. First, it would be helpful to see the measurement uncertainties on this plot, as the data aren't super convincingly supportive of a maxima. Second, to suggest that there was "no subsurface maxima" in Figure 4(c) when only two measurements were made between the surface and the 25 m depth suggests that it is entirely possible that there is a subsurface maxima that just wasn't observed because no 20 m depth was measured. This needs to be included in the discussion.

Page 16, lines 18-20 – it would be nice to see a consideration of the daytime/nighttime differences in the correlations – if, as the authors are suggesting, there is atmospheric accumulation during the night, one might expect a better correlation during the night than during the day.

Page 17, line 29 – ". . . in the latter case."

Page 21, line 21 – are there really no units for salinity?

Page 29, line 2 – "note the colorbar in. . ."

Page 29, line 5 – I don't think "Global" is necessarily the right adjective of the observed radiation. Also, for this plot (Figure 3) and Figure 6, it would be better to change either the black or blue dots to a slightly different color, because they look very similar. Perhaps change the symbols, too, so that they're not all circles.

---

## Author Comment (AC1) · 10 Jun 2016

General comments: This manuscript by Hepach et al. reported their observations on a series of bromo- and iodocarbons from surface ocean as well as subsurface waters in the Peruvian upwelling zone. The authors further investigated into possible sources of the halocarbons and the contribution of organoiodine to stratosphere iodine loading. This manuscript is relatively well written and discussed drivers for the production of the halocarbons and transport of iodinated species, in addition to reporting data. I believe it is a new contribution to the scientific community. The authors also proposed directions for future studies for addressing the halocarbon budgets. I believe this manuscript is suitable for publication in ACP with minor revisions. Please see specific comments

suggested below.

- We thank referee #1 for the review. We will address the specific comments in the following. All corrections according to referee #1 will be marked in blue in the manuscript. Further changes are marked in green.

Specific comments Page 2 Line 2: delete "halogenated short-chained hydrocarbons" for "halocarbons" is clear enough for this definition.

- Done.

Page 2 Line 6: should read "Peruvian upwelling zone (or regions)".

- We added "zone".

Page 2 Line 23: perhaps use "very short-lived substances (VLS)" following the WMO terminology?

- Since we use "halocarbons" as terminology throughout the manuscript, we decide to keep it. The terminology is used in the oceanographic and lower troposphere community, while VLS also include compounds without halogens, thus we decided to be more specific with this term. To avoid confusion, we delete "organic compounds" and write "halocarbons" instead.

Page 3 Line 16: please clarify this sentence by adding "in seawater" after "for both compounds".

- We added "in the surface ocean" to clarify the location for which this is valid.

Page 3 Line 29: should read "the main sinks for both $CH_2I_2$ and $CH_2ClI$ are"

- Since it is only one sink, this should be singular. But we agree that it may be written a bit confusing, so we rewrite it as "The main sink for both $CH_2I_2$ and $CH_2ClI$ is photolytical breakdown [. . .]".

Page 4 Line 14: replace "the latter two compounds" with "$CH_2ClI$ and $CH_2I_2$" to avoid

confusion.

- Done.

Page 5 Line 2: it is a bit confusing here, is it "every 3 hours"?

- Yes, we clarified this sentence. See also changes with respect to referee #2.

Page 5 Line 6 and 7: were purge efficiency measured for these gases?

- Purge efficiencies were determined in previous laboratory experiments. We purge 50 mL of water at 70 °C with a stream of helium of 30 mL min-1 for 50 min. With these conditions, we achieve a purge efficiency of larger than 98 % for all five compounds. We add a short sentence with these specifications.

Page 5 Line 11: please specified how the gas samples were stored – stainless steel canisters?

- Yes, the gas samples were taken in pre-cleaned stainless steel canisters. This is described in Fuhlbrügge et al. (2015) in more detail, but we also add this to this part of the manuscript.

Page 5 Line 25: Cyanobacteria may pass through the GF/F filters at the initial filtering (i.e. before the filter pore size decreased as materials accumulated), which may affect the quantification of the cyanobacteria marker pigments. Did the authors estimate such a biomass lost?

- The referee is correct that concern has been raised that filtering through GF/F filters may sometimes cause very small cyanobacteria, mainly prochlorophytes, to pass through the filter. The smallest prokaryotic algae are probably not measured with the same accuracy as all other phytoplankton cells above 0.7 $\mu$m by this technique (Dickson and Wheeler 1993). However, Chavez et al. (1995) compared HPLC results from GF/F 0.7 $\mu$m and membrane 0.2 $\mu$m filters, and clearly showed that GF/F filters can be used to accurately measure pigments also from very small phytoplankton. Less

than 1 % of prochlorophytes pass through the GF/F filters. We believe that the error caused by some of the smallest prokaryotic algae passing through the filter and not being accurately accounted for, is very low for our cruise.

Page 5 Lines 28 to Page 6 Lines 1 to 7: The DOM samples were collected from 20 cm and the gases were collected from about 6 to 7 m, which were not exactly parallel samples. Some DOM can be recycled relatively fast. In addition, DOM at surface ocean may be degraded via photolysis. I suggest the authors to also report the mix layer depth and possible residence times for the DOM compounds they measured, such that a valid argument can be made about those DOM were well mixed within the mixed layer and hence the depth difference would not affect the data analysis and interpretation.

- The mixed layer depths during M91 were rather shallow, usually between 6 and 25 m. We assume that this very upper layer was well mixed, so samples for both components were taken from the same water masses. Unpublished data from a recent cruise (ASTRA-OMZ, SO243 onboard the RV Sonne) show that our halocarbon measurements from the first meter are in good agreement with measurements from the hydrographic shaft if the mixed layer is not shallower than the shaft depth. High molecular weight dissolved combined carbohydrates, such as determined during this study, have been considered as labile to semi-labile with turn-over times of several days to months (Engel et al. 2011; Hansell, 2013), suggesting that the mixing time in the upper water is faster than the turn-over of the combined sugars, which will also be added to the manuscript. We add in the methods section: "Very-well mixed layers at these measurement locations reach down to between 6 and 25 m, and DOM turn-over times for the respective compounds has been reported to be several days to months (Engel et al., 2011; Hansell, 2013)."

Page 8 Lines 1 to 15: I suggest move this to before the FLEXPART model simulation.

- Done.

[Figure]

Page 9 Line 2: change "correlated very well" to "significantly correlated".

- Done.

Page 11 Line 11: I suggest the authors also include the depth profile of the bromocarbons.

- We added the bromocarbons to Fig. 4. The text in the manuscript is changed accordingly (section 4.2 and the figure caption).

Page 11 Lines 22 to 24: this sentence is a bit confusing, please rephrase.

- We agree and rewrite as "$CH_2I_2$ was hardly detected in deeper water in the northern part of our measurements (Figure 4, upper panel)."

Page 12 Line 22: Liu et al., 2015 tested a series of carbohydrates, and found that these DOM moieties were not fast reacting substrates for $CHBr_3$, which seems to be consistent with findings in this study. In addition, bromocarbon formations via HOBr reaction are potentially DOM moiety specific. In Liu et al., 2013, no correlations were observed between the bromocarbons and total dissolved organic carbon.

- Thank you for this suggestion. We include this hypothesis in the manuscript and write: "Bromocarbon production from DOM has also been suggested to be slow (Liu et al. 2015), which could shift larger bromocarbon concentrations to later times after our cruise."

Page 13 Line 12: the bulk DOM may correlate better with total biomass (estimated from TChla).

- This is certainly a reasonable assumption. However, we tested the data and correlations are not much better. This may be due to the fact that the main part of the total biomass is made up of diatoms. Hence, the correlations between the total biomass and DOM are mainly regulated by diatom abundance, which leads to similar correlations for biomass and DOM vs diatoms and DOM.

Page 13 Lines13 to 15: I suggest change "is determined by the phytoplankton species" to "is determined by ecosystem compositions", because DOM contribution is not governed by phytoplankton species alone.

- Done.

Page 13 Line 21: Please also cite Lin and Manley 2012, who also tested bromocarbon formations using different molecular weight natural DOM as substrates.

- Done.

Page 13 Lines 24 to 27: The authors depicted possible abiotic sources of HOI and HOBr in Fig 5. I would suggest the authors also put the abiotic sources of HOI and HOBr into this context (see Carpenter et al., 2005).

- We included this in the figure, but only schematically, since we focus on the biotic formation in our manuscript.

Page 14 Line 6: I suggest remove "In conclusion" here.

- Done.

Page 31 Fig 5: The conception model figure is a bit confusing on the CH3I part via methyltransferase, for it is an intracellular enzyme. Thus the reaction is likely occurring inside the cell. However, figure seems to depict an extracellular reaction.

- We agree, and it was actually intended as such. Thus, we make this clearer in the figure now.

References Chavez, F. P., Buck, K.R., Bidigare, R.R., Karl, D.M., Hebel, D.V., Latasa, M., Campbell, L., and Newton, J.: On the chlorophyll-a retention properties of glass-fiber GF/F filters, Limnol. Oceanogr., 40, 428-433, 10.4319/lo.1995.40.2.0428, 1995. Dickson, M.-L., and Wheeler P. A.: Chlorophyll a concentrations in the North Pacific: Does a latitudinal gradient exist?, Limnol. Oceanogr., 38, 1813-l818, 10.4319/lo.1993.38.8.1813, 1993. Engel, A., Händel, N., Wohlers, J., Lunau, M.,

Grossart, H. P., Sommer, U. und Riebesell, U.: Effects of sea surface warming on the production and composition of dissolved organic matter during phytoplankton blooms: results from a mesocosm study, J. Plankton Res., 33, 357-372, 10.1093/plankt/fbq122, 2011. Fuhlbrügge, S., Quack, B., Atlas, E., Fiehn, A., Hepach, H., and Krüger, K.: Meteorological constraints on oceanic halocarbons above the peruvian upwelling, Atmos. Chem. Phys. Discuss., 15, 20597-20628, 10.5194/acpd-15-20597-2015, 2015. Hansell, D. A.: Recalcitrant dissolved organic carbon fractions, Ann. Rev. Mar. Sci., 5, 421-445, 10.1146/annurev-marine-120710-100757, 2013. Marie, D., Simon, N., and Vaulot, D.: Phytoplankton cell counting by flow cytometry, in: Algal Culturing Techniques, edited by Andersen, R. A., Elsevier Academic Press, 2005. Taylor B.B., Torrecilla, E., Bernhardt, A., Taylor, M. H., Peeken, I., Röttgers, R., Piera, J., and Bracher, A.: Bio-optical provinces in the eastern Atlantic Ocean, Biogeosciences, 8, 3609-3629, 10.5194/bg-8-3609-2011,

---

## Author Comment (AC2) · 10 Jun 2016

General Comments This manuscript by Hepach and co-authors presents gas-phase and oceanic observations of halogenated VOCs, including CHBr3, CH2Br2, CH2ClI, CH3I, and CH2I2 from a ship cruise in the eastern tropical Pacific Ocean. In addition to the concentrations and sea-air flux calculations, the analysis includes correlations to phytoplankton groups measured in the surface water along the cruise path. The paper is reasonably well written and the many of the data presented are new observations. The paper should be published in ACP after addressing the following minor corrections.

- We thank referee #2 for this review. We will address the specific comments in the following. All corrections with respect to the suggestions by referee #2 will be marked

in red in the manuscript. Further changes are marked in green.

Specific Comments Page 5, line 2 – Three hourly is not explicit – every three hours, or three samples per hour? And was this day and night? What were the samples taken in? I would like to see a little more detail, even though the system was described in another paper.

- Samples were taken every three hours throughout the whole day, hence both during day and night, adding up to in total eight samples. Water was sampled in amber glass bottles. We added respective details in the manuscript. See also our changes with respect to referee #1.

Page 5, line 7 – replace "lay" with "was".

- Done.

Page 5, line 9 – "set up problems" sounds odd. Perhaps "instrument issues?"

- We replace this with the suggestion of the referee.

Page 5, line 10 – 20 m "above sea level?" How were these samples stored? Stainless canisters? Glass flasks?

- We added "above sea level" to the section. For the second issue, please see our answer to referee #1.

Page 5, line 11 – "starting on December 1"

- Done.

Page 5, line 23 – replace "build up" with "comprise"

- Done.

Page 5, line 26 – "a" should be italicized.

- Agreed.

Page 7, line 20 – The authors explain that they do not include detailed tropospheric iodine chemistry, and specify what they, explicit removal of HOI, HI, IONO2, and IxOy through scavenging or 20 heterogeneous recycling of HOI, IONO2, and INO2 on aerosols, and then reference Saiz-Lopez et al., 2014. It should be made clear whether or not Saiz-Lopez did or did not omit iodine removal . I.e., "we didn't do x, y and z (reference)" - is the reference an example where x, y and z were omitted, or not omitted?

- Thanks for pointing this out. Since the Saiz-Lopez et al., 2014 study includes detailed tropospheric iodine chemistry and removal, it is not an appropriate reference in this sentence and has been removed in the current version of the manuscript.

Page 8 – section 2.6 and section 2.5 should be swapped (i.e., measurement methods before model description.)

- We agree. See changes according to referee #1.

Page 10, lines 1-4 – The suggestion that the observations from this work "compare well" with observations from Liu et al. (2013) needs to be backed up with something more quantitative. CHBr3 seems to have a similar range, but the CH2Br2 range from Liu et al. are about half the values from the current work. Can you be more specific about the region covered by Liu et al., i.e. where the observed maxima were located?

- The campaign covered a large region between Punta Arenas, Chile and Seattle, USA. The coastal region covered by Liu et al. (2013) was south of 40° S, while we investigated a region between 5° S and 16° S, where the data of Liu et al. (2013) only consist of open ocean data, which could explain the difference in the ranges between our study and theirs. Unfortunately, not much more data exist for the coastal tropical East Pacific. However, Liu et al. (2013) measured about 20 pmol L-1 of CHBr3 and around 6 pmol L-1 of CH2Br2, showing that also the more southern part of the South American coast is characterized by concentrations in our range. We will specify this in the manuscript.

Page 11, line 11 – "during a large part"

- Done.

Page 11, lines 18-21 – I have a bit of an issue with this interpretation. In Figure 4, we see that there were four subsurface measurements made between the surface and a depth of 25 m, and one single subsurface maxima is shown at a depth of about 20. First, it would be helpful to see the measurement uncertainties on this plot, as the data aren't super convincingly supportive of a maximum. Second, to suggest that there was "no subsurface maxima" in Figure 4(c) when only two measurements were made between the surface and the 25 m depth suggests that it is entirely possible that there is a subsurface maxima that just wasn't observed because no 20 m depth was measured. This needs to be included in the discussion.

- The measurement error is estimated to be around 10 % for all halocarbon measurements in seawater. The maximum concentrations shown are well outside of this 10 % range. We agree that there could be a maximum at 20 m. We do not preclude a subsurface maximum per se at this particular profile, we merely state that we observed it considerably often in the mentioned upwelling region. This profile was chosen, because the mentioned features could be observed for all iodocarbons. We clarify this in the manuscript.

Page 16, lines 18-20 – it would be nice to see a consideration of the daytime/nighttime differences in the correlations – if, as the authors are suggesting, there is atmospheric accumulation during the night, one might expect a better correlation during the night than during the day.

- Thank you for this suggestion. We tested this hypothesis by dividing the data into "day" and "night" by assuming "night" was when there was no radiation at all. Then, we correlated the data again. The shorter the lifetime of the compound, the better is the correlations between the oceanic and atmospheric halocarbons. Considering only data points when there is light, oceanic and atmospheric $CH_2ClI$ correlate significantly

with r = 0.5, during the night, this correlation is stronger with r = 0.68. This observation is even clearer for CH2I2, which correlates not significantly during the "day" with r = 0.4, owing to the very few data points during light hours, and very strongly during the night with r = 0.92. This supports our hypothesis that atmospheric CH2ClI and CH2I2 build up during the night, and are degraded during the day. We include this separation in the discussion, and add a figure depicting this (new Fig. 7).

Page 17, line 29 – "in the latter case."

- Done.

Page 21, line 21 – are there really no units for salinity?

- Salinity measurements here are based on the conductivity ratio, which is dimensionless. The use of the commonly applied unit "PSU" (Practical Salinity Unit) is not recommended anymore, hence, no units are used here. For more information see Millero (2010).

Page 29, line 2 – "note the colorbar in"

- Changed.

Page 29, line 5 – I don't think "Global" is necessarily the right adjective of the observed radiation. Also, for this plot (Figure 3) and Figure 6, it would be better to change either the black or blue dots to a slightly different color, because they look very similar. Perhaps change the symbols, too, so that they're not all circles.

- "Global radiation" is a parameter measured onboard by a pyranometer and refers to the total short-wave radiation, which includes both the direct solar radiation and the diffuse radiation. Hence, the adjective hereby is part of the parameter name. This is made clearer in the figure description now. The blue symbols were changed to a lighter color and the line style was changed for better distinction.

References  Liu, Y. N., Yvon-Lewis, S. A., Thornton, D. C. O., Campbell, L., and

Bianchi, T. S.: Spatial distribution of brominated very short-lived substances in the eastern pacific, J. Geophys. Res.-Oceans, 118, 2318-2328, 10.1002/jgrc.20183, 2013.

Millero, F. J.: History of the equation of state of seawater, Oceanography, 23, 18-33, 10.5670/oceanog.2010.21, 2010.
* * *

---

## Author Response (AR2)

**Comments to the Author:**

The revised manuscript has improved significantly. Most of the comments by the reviewers have been adequately addressed. However, the summary of previous measurements over the

5 Eastern Pacific ocean is incomplete. There is several recent experimental studies from ship and aircraft, and modeling studies that integrate organic and inorganic iodine sources from the ocean surface, and into the stratosphere. These recent literature should be added, and discussed in context of the results presented here before the manuscript is ready for publication.

10

30

We thank the editor for his suggestions and comments, and hope that we have improved the manuscript accordingly. All corrections are marked in red in the manuscript below. We have added more data comparisons with focus on the tropical East Pacific especially in the atmospheric discussion sections. As we concentrate mainly on the MABL in our manuscript,

15 we have chosen to mainly compare our data with MABL measurements as well.

Reviewer#1 has suggested to depicted "abiotic sources of HOI and HOBr in Fig 5". The Editor agrees with this suggestion. While the formation of VSLs is the primary focus of this paper, HOI is arguably the largest source of iodine from the ocean (Sherwen et al. 2016).

20 Fig. 5 is missing direct emissions of HOI, and should be revised to reflect the state-of-thescience, rather than the scope of this paper. This will broaden the appeal of the Figure & paper.

We added the inorganic iodine cycling via iodide and ozone schematically to the figure, and hope that it reflects better current knowledge.

Reviewer#2 points out that the authors do not explicitly include tropospheric iodine chemistry in their model. The authors circumvent this by sensitivity studies that vary the lifetime of iodine, which is fine, but no reference is given to support the lifetime. Please provide references in support of the iodine lifetime.

We have added a reference (Sherwen et al., 2016) to the manuscript.

With respect to the discussion of iodine injections to the stratosphere: the contribution of CH3I was recently compared with inorganic iodine injections (Saiz Lopez et al. 2015). Discussion is missing about how the CH3I data presented by the authors compare with these estimates, and in context of the relevance of inorganic iodine sources to the stratosphere.

- 5 We have added a short discussion of such a comparison. Note however, that this is not straightforward and somewhat speculative since neither Saiz-Lopez et al. (2015) nor Ordóñez et al. (2012) show their CH3I emissions or boundary layer concentrations. Our paper on the other hand only provides boundary layer estimates and does not include any simulations of stratospheric contributions.
- 10

The statement "that for the Peruvian upwelling region... higher iodocarbon sources lead to larger IO abundances" is partially supported by Fig. 8. There is equal evidence to the conrtary in that Figure. The statement that "a considerable part of the atmospheric IO variations [can be explained] with the variability of the oceanic organic sources" does not

- 15 seem well supported. Which variability in IO are the authors referring to in Fig. 8b? A correlation plot should be added in support of this statement, or the language be modified. In the present form there is equal evidence in Fig. 8b to support that the contribution of combined iodocarbon fluxes to IO can be as low as few %. The manuscript language (abstract, discussion, conclusion) should be adjusted to reflect all data presented in this
- 20 paper. At present it seems somewhat biased to conditions when the contributions of organic iodine fluxes were significant.

25

We have changed abstract, discussion and conclusion as requested by the editor to make clearer that during the second part of the cruise the contribution of the iodocarbon sources to atmospheric IO is very small. Additionally, we toned down the sentence referring to the variations of IO vs the variability of organic sources.

30

**Biogenic halocarbons from the Peruvian upwelling region as tropospheric halogen source**

Helmke Hepach1,6, Birgit Quack1, Susann Tegtmeier1, Anja Engel1, Astrid
Bracher2, Steffen Fuhlbrügge1, Luisa Galgani1,7, Elliot L. Atlas3, Johannes Lampel4, Udo Frieß4, and Kirstin Krüger5

[revised manuscript text omitted]